# Comparative performance of different methods for circulating tumor cell enrichment in metastatic breast cancer patients

Arik Drucker[1], Evelyn M. Teh[2], Ripsik Kostyleva[2], Daniel Rayson[1], Susan Douglas[2], Devanand M. Pinto[2]*

1 Division of Medical Oncology, Department of Medicine, Dalhousie University and Nova Scotia Health Authority, Halifax, Nova Scotia, Canada, 2 Human Health Therapeutics Research Centre, National Research Council of Canada, Halifax, Nova Scotia, Canada

* Devanand.Pinto@nrc-cnrc.gc.ca

**Data Availability Statement:** Anonymized raw data used for generating box plot figures is provided in the Supporting Information files.

## Abstract

The isolation and analysis of circulating tumor cells (CTC) has the potential to provide minimally invasive diagnostic, prognostic and predictive information. Widespread clinical implementation of CTC analysis has been hampered by a lack of comparative investigation between different analytic methodologies in clinically relevant settings. The objective of this study was to evaluate four different CTC isolation techniques–those that rely on surface antigen expression (EpCAM or CD45 using DynaBeads® or EasySep™ systems) or the biophysical properties (RosetteSep™ or ScreenCell®) of CTCs. These were evaluated using cultured cells in order to calculate isolation efficiency at various levels including; inter-assay and inter-operator variability, protocol complexity and turn-around time. All four techniques were adequate at levels above 100 cells/mL which is commonly used for the evaluation of new isolation techniques. Only the RosetteSep™ and ScreenCell® techniques were found to provide adequate sensitivity at a level of 10 cells/mL. These techniques were then applied to the isolation and analysis of circulating tumor cells blood drawn from metastatic breast cancer patients where CTCs were detected in 54% (15/28) of MBC patients using the RosetteSep™ and 75% (6/8) with ScreenCell®. Overall, the ScreenCell® method had better sensitivity.

## Introduction

Dissemination of tumor cells from primary tumors to distant sites is a principal cause of cancer-specific mortality. As such, a better understanding of the metastatic process is a critical step in improving survival outcomes. The isolation and molecular characterization of circulating tumor cells (CTCs) is an active area of research and has improved our understanding of the mechanisms of metastasis [1–5]. Although intriguing, these results are preliminary and require validation in larger patient cohorts. However, such research is constrained by technical challenges in CTC isolation, particularly those that arise from difficulties in identifying and separating blood-borne CTCs from a large background of normal blood cells, which outnumber CTCs by over a million-fold.

**Funding:** A.D - University Internal Medicine Research Fund, CDHA Category 2 Fund, Pfizer Oncology IIR D.P., S.D., E.T.- National Research Council Atlantic Initiative The funders had no role in study design, data collection and analysis, decision to publish, or preparation of the manuscript.

**Competing interests:** Funding for this study was provided in part from a Pfizer Investigator Initiated Research Grant (WS575460). This does not alter our adherence to PLOS ONE policies on sharing data and materials. We have no further relevant declarations (such as consultancies, patents, employment, product development) related to this funder or any other commercial entities.

CTC isolation methodologies can be grouped into one of two categories: separation based on physical properties such as size or deformability and selection via expression of specific markers, most commonly epithelial cell adhesion molecule (EpCAM). The scarcity of CTCs requires CTC isolation techniques that have both exceptionally high sensitivity and high specificity. Although numerous techniques have been developed for CTC isolation (as recently reviewed [6,7]), only the CellSearch® technology has received FDA regulatory approval for measurement of CTC in clinical practice. This system relies on positive selection of CTCs using anti-EpCAM antibodies and has shown clinical utility for breast, prostate, and colon cancer [8–10]. However, positive selection techniques are less effective for CTC analysis of other solid tumors with low EpCAM expression, most notably non-small cell lung cancer and triple-negative (i.e., estrogen and progesterone receptor-negative, and HER2/neu-negative) breast cancer, which have a higher metastatic potential, and poorer survival outcomes compared to other breast cancer subtypes [11]. There is a need to develop technologies that are more broadly applicable to a variety of cancer types, particularly because of the relative ease of CTC collection compared to conventional invasive tissue biopsy.

The development of novel methods for CTC isolation is an active area of research; several dozen techniques for CTC isolation have been published in the past 5 years. These technologies promise enormous improvements in oncology, such as dynamic monitoring of therapeutic effect [8], reduced reliance on imaging for treatment decision making [12], and tailoring systemic therapy to identifiable molecular aberrations [13, 14]. However, most studies focus on development of novel methods for CTC capture and make use of cultured cells spiked into normal blood. Very few studies report benchmarking or objective comparison of several techniques and even fewer studies compare techniques in a clinical setting. This study evaluates the performance of four CTC isolation techniques that are based on the expression of the cell surface antigen, EpCAM or the biophysical properties such as size and density of CTCs. Both the Dynabeads® (ThermoFisher Scientific, USA) and EasySep™ (StemCell™ Technologies, Vancouver, Canada) are immunomagnetic-based methods where antibodies recognizing cell surface antigens are coupled to magnetic beads and used to either remove unwanted cells (negative depletion) or enrich CTCs (positive selection). By contrast, the RosetteSep™ (StemCell™ Technologies) system combines the use of antibodies to change the density of unwanted cells followed by subsequent removal by density gradient centrifugation. Finally, ScreenCell® (Paris, France) uses micro-filters to separate the larger CTCs from leukocytes.

The objective of this study is to find a simple and rapid method that provides sufficient sensitivity, recovery, and specificity in CTC identification. Techniques that satisfy these criteria would enable more widespread use of CTC analysis in clinical research relative to more complex and costly approaches to CTC isolation. The methods listed above were evaluated using a known amount of cultured cells spiked into blood from healthy volunteers in order to calculate figures of merit such as recovery and reproducibility. Two promising techniques were then evaluated for CTC analysis in metastatic breast cancer patients in a head-to-head comparison.

## Materials and methods

### Breast cancer cell line

The MDA-MB-231 cell line used in the spiking experiments were purchased from ATCC (Manassas, VA). Cells were maintained in Dulbecco's Minimum Essential Media (DMEM; ThermoFisher Scientific, USA) supplemented with 10% fetal bovine serum (FBS, ThermoFisher Scientific). All cells were grown in a humidified incubator at 37˚C with 5% $CO_2$.

## Spiking experiments with healthy controls

Peripheral blood was drawn from healthy volunteers according to the study protocol, which was approved by the Capital Health Research Ethics Board (Study Identifier: CDHA-RS/2009-088), Halifax, Nova Scotia and the National Research Council of Canada (2009–10). All participants signed an informed consent prior to enrolment. Approximately 20 mL of blood were drawn from healthy volunteers into EDTA vacutainers (BD, New Jersey, USA). Healthy volunteers did not have any prior or current malignancy or autoimmune diseases. Known quantities of MDA-MB-231 cells were spiked into whole blood from healthy volunteers, subjected to target cell enrichments, and enumerated by immunocytochemistry (ICC). Spiking experiments were performed at least three times on separate days to evaluate reproducibility. For spike-in samples containing 100 to $10^3$ cells, serial dilution of a stock solution was used. For samples containing fewer than 10 cells, manual micro-pipetting with the aid of a microscope was used. Un-spiked samples were used as negative controls.

## Target cell enrichment using cell surface antigen expression

Isolation efficiency of spiked cultured cells into whole blood was first performed as shown in Fig 1A using negative depletion (leukocyte depletion) coupled to positive selection (target cell enrichment). Cultured cells were harvested by trypsinization, washed in phosphate buffered saline (PBS), and enumerated using the Countess Automated Cell counter (ThermoFisher Scientific). Cells were serially diluted or manually pipetted into PBS containing 2% FBS (PBS + 2% FBS) to represent approximately 1000, 100, 10 per 100 μL. The cells were spiked into 5 mL of fresh whole blood from healthy volunteers. Mononuclear cells were isolated by diluting whole blood 1:1 with PBS +2% FBS and careful layering onto Ficoll-Paque™ Plus (GE Healthcare, Mississauga, Canada) followed by centrifugation at 400g for 30 min at room temperature.

Leukocytes were first depleted from the cell mixture using magnetic beads coupled to anti-CD45 (Dynabeads® CD45 or EasySep™ Human CD45 Depletion Kit) followed by enrichment

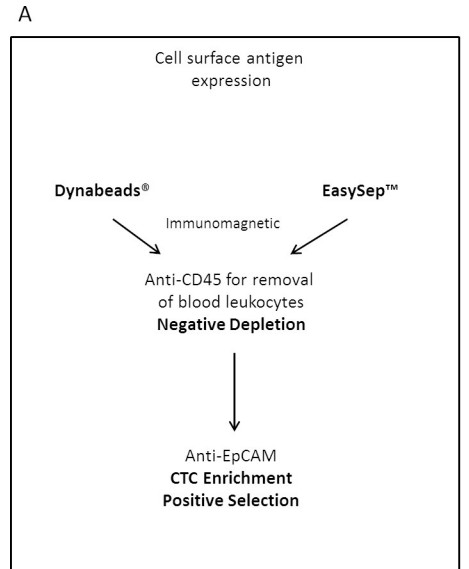
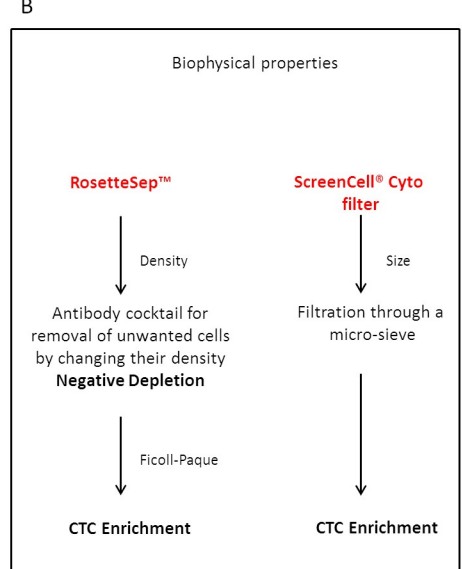
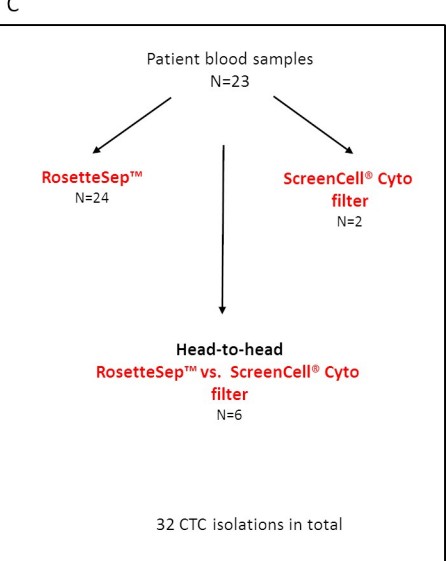

**Fig 1. Summary of methodologies used for CTC enrichment.** Cultured cells were spiked into whole blood to test the recovery efficiency using the different methodologies. (A) The Dynabeads® and EasySep™ methods rely on the cell surface antigen expression were respectively used first to perform a negative depletion followed by CTC enrichment. (B) The RosetteSep™ and ScreenCell® Cyto filters were used to evaluate the CTC capture efficiency based on biophysical properties of the tumor cell. (C) The two methodologies, RosetteSep™ and ScreenCell®, were further used on patient blood samples for CTC enrichment. Four patients had multiple sampling during the course of disease progression. Six patients were isolated with both methodologies in a head-to-head comparison.

with anti-EpCAM (Dynabeads® Epithelial Enrich or EasySep™ Human EpCAM Positive Selection Kit) (Fig 1A). Isolations were performed according to the manufacturer's instructions with modifications in the final step. Specifically, after negative selection by anti-CD45 the supernatant was further subjected to immunomagnetic positive selection using anti-EpCAM coupled to either DynaBeads® or the EasySep™ system. Recovered cells were resuspended in 100 μL of PBS + 2% FBS and cytospins prepared using the StatSpin Cytofuge (ThermoFisher Scientific) onto 3-triethoxysilylpropylamine (TESPA)-coated glass slides. Slides were dried overnight before fixation and permeabilization in pre-chilled (-20˚C) methanol for 10 min followed by incubation in pre-chilled acetone for 1 min. The slides were dried briefly before storing at -20˚C. The cytospins were used for identification and enumeration of CTCs by immunocytochemistry (ICC).

## Target cell enrichment using biophysical methods

An immunodensity method using the RosetteSep™ Human Circulating Epithelial Tumor Cell Enrichment Cocktail (StemCell™ Technologies) combined with the Ficoll-gradient centrifugation was also evaluated (Fig 1B). Spiking of cultured cells into blood was performed as above and the isolation was performed as per the manufacturer's instructions. Briefly, 250 μL (50 μL/mL) of the RosetteSep™ cocktail was added to the 5 mL of spiked blood and incubated for 20 min at room temperature. Blood samples were then diluted with equal volumes of PBS + 2% FBS and layered carefully onto Ficoll-Paque™ Plus and centrifuged at 1,200g at room temperature for 20 min for separation. The enriched cells in the Ficoll: plasma interface were collected, residual red blood cells lysed by $NH_4Cl$ buffer (154 mM $NH_4Cl$, 10 mM $KHCO_3$, 0.1 mM EDTA) and washed in PBS + 2% FBS. Cells were resuspended in 100 μL of PBS + 2% FBS after the final wash and cytospins prepared for target cell enumeration. The RosetteSep™ method was also combined with the EasySep™ Human EpCAM Positive Selection Kit to achieve higher cell purity. The recovered cells from the RosetteSep™ depletion was used for target cell enrichment with the EasySep™ EpCAM nanoparticles.

The RosetteSep™ method was further evaluated for inter-assay variability at 100 and 10 cells by processing by 3 separate operators in three experiments conducted on two separate days. The inter-observer variability in data interpretation. In this case, two observers viewed and scored the same cyctospin samples for CTC enumeration. The RosetteSep™ method was subsequently used for CTC isolations from 21 metastatic breast cancer patients. Cytospins were prepared from the recovered cells for identification and enumeration by ICC.

During this study, the ScreenCell® Cyto kits (ScreenCell®, Paris, France) which are non-invasive blood filters used for CTC enrichment by size became available. These filter devices consist of random pores of 7.5 μm and uses a 10 mL blood vacutainer to suction blood through the filter from the upper chamber [15] (Fig 1B). Cultured MDA-MB-231 cells were spiked at 100 cells into 3 mL of whole blood for evaluation of recovery. The ScreenCell® Cyto kits were used for CTC enumeration and cytomorphological analysis in 8 metastatic breast cancer patients. CTC enrichment using ScreenCell® devices was performed according to the manufacturer's protocol. Blood was incubated with the provided formaldehyde-based (FC2) buffer for 8 min at RT and subsequently filtered through the Cyto devices. The filters were dried before storing at -20˚C.

## Patient sampling and blood processing

Peripheral blood (15 mL) was collected from 23 recurrent or newly diagnosed metastatic breast cancer patients. Blood was drawn and stored in EDTA vacutainer tubes. Patient samples were collected between September 2010 –September 2013. The eligibility criteria were age over 18 years, confirmed diagnosis of metastatic breast cancer and active follow up at the Nova Scotia Cancer Centre. The study protocol was approved by the Capital Health Research Ethics Board

(Study Identifier: CDHA-RS/2009-088), Halifax, Nova Scotia and the National Research Council of Canada (2009–10).

Blood was processed within 4 hours of collection. To avoid possible contamination from skin cells during venipuncture, the first tube of blood was not used for CTC isolation. From subsequent tubes, between 3–5 mL blood was used for CTC enrichment using either the RosetteSep™ method and/or by ScreenCell® Cyto kits (Fig 1C)

## Identification and enumeration by Immunocytochemistry (ICC)

Immunocytochemistry was performed on cytospins and ScreenCell® filters by indirect labelling using a mouse monoclonal antibody against human pan cytokeratins (AE1/AE3 + 5D3; Abcam, Cambridge, USA), mouse monoclonal anti-human CK-7 (RCK105; Abcam), mouse monoclonal anti-human vimentin (V9; Abcam) and a rabbit polyclonal antibody against human pan-CD45 (Abcam). Goat anti-mouse and goat anti-rabbit secondary antibodies coupled to Alexa Fluor® 488 (ThermoFisher) and Alexa Fluor® 568 (ThermoFisher) were used to differentiate leukocytes from CTCs. 4', 6' - diamidino-2-phenylindole (DAPI) was used to stain the nucleus.

Cytospins and formaldehyde-fixed cells on the ScreenCell® Cyto filters were rehydrated in PBS. ScreenCell® Cyto filters were also subjected to antigen retrieval in 10 mM sodium citrate, pH 6 + 0.005% Tween 20 (Millipore Sigma, Oakville, Canada) at 95°C for 20 min. After cooling, cell permeabilization was performed in PBS containing 0.2% Triton X-100 (Millipore Sigma) for 5 min at room temperature. A 10% goat serum (ThermoFisher) in PBS + 0.05% Tween-20 was used as the blocking solution for both cytospins and ScreenCell® Cyto filters. A 1/250 dilution was used for the primary antibodies anti-pan CK/CK-7 and anti-CD45 in both the cytospins and ScreenCell® Cyto filters for 1 h at room temperature in a humidified chamber. In some slides, anti-vimentin at 1/500 dilution was also included in the primary antibody incubation. Slides and filters were washed 3X in PBS before incubation in the secondary antibodies (1/1000; goat anti-mouse Alexa Fluor® 488 and goat anti rabbit Alexa Fluor® 568) for 1 h at room temperature in a humidified chamber. After the final washes in PBS, the slides and filters were rinsed briefly in water before the application of Fluoro Gel II with DAPI (Electron Microscopy Sciences, Hatfield, USA) and cover-slipped. Cells were viewed under a Leica DMRE fluorescence microscope using a 40X objective lens equipped with a Hamamatsu 1394 ORCA-285 camera and the SimplePCI software (Compix Inc., Sewickley, USA). Images were pseudo-colored and merged using Image J (Version 1.4r) [16]. Cells with pan-CK/CK-7$^+$/ CD45$^-$ and an intact nucleus with DAPI staining were defined as a CTC and counted manually. In addition to the staining criteria, cells were also evaluated based on the cell size ($> 5$ μm) and cyto-morphological appearance such as enlarged nuclear size with high nuclear to cytoplasmic ratio, especially when CK staining was variable. Positive controls consisting of 100 or 500 spiked MDA-MB-231 cells into healthy blood on cytospins were included in each experiment.

## Statistical analysis

Experimental replicates for the cell spiking experiments are reported as mean ± SD. Unpaired, two tailed T-test of unequal variance was performed in Excel. ANOVA and statistical analysis for the inter-assay and inter-observer analysis were performed using R (version 3.0.1). A $p$ value of less than 0.05 was used to indicate a significant difference.

## Results

### Enrichment assay validation and optimization

Two different immunomagnetic systems, Dynabeads® and EasySep™ for target cell enrichment were first tested. Since high numbers of contaminating leukocytes have been reported

when using either negative depletion [17] or positive cell enrichment [15, 18] alone, we assessed the benefits of combining negative depletion followed by positive enrichment to reduce the number of contaminating leukocytes. The efficacy of the combined method in first removing the unwanted cells followed by enrichment of target cells was evaluated using cultured breast cancer cells spiked into 5 mL of whole blood. Of the two immunomagnetic systems for combined negative followed by positive selection, the combined Dynabead® (n = 5) system was more efficient in target cell recoveries with a mean of 44 ± 23% compared to the EasySep™ system (n = 6) which had a mean recovery of 24% ± 19% when cultured cells were spiked in at $10^4$. When a 1000 cell spike-in was performed, the mean recovery for the Dynabeads® (n = 5) fell to 9% ± 6% while the EasySep™ (n = 3) had a mean recovery of 2% ± 2%. While leukocyte contamination appeared to be reduced in the combined methods compared to either negative or positive selection alone, the poor recovery rates did not warrant the continuation of this methodology.

To compare the immunomagnetic based cell isolations with the non-immunomagnetic techniques, the RosetteSep™ Human Circulating Epithelial Tumor Cell Enrichment Cocktail and ScreenCell® filters were also assessed. The RosetteSep™ method relies on changing the density of unwanted cells with antibodies followed by centrifugation with Ficoll-Paque for target cell enrichment. The RosetteSep™ method (n = 7) gave an isolation recovery with a mean value of 34% ± 15% in the 1000 cell spike-ins, which was better (p = 0.004) than both the immunomagnetic-based techniques (Fig 2A). Spike-ins of cultured cells at 100 (n = 6) and 10

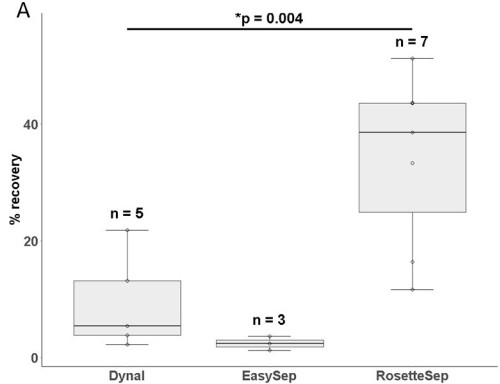

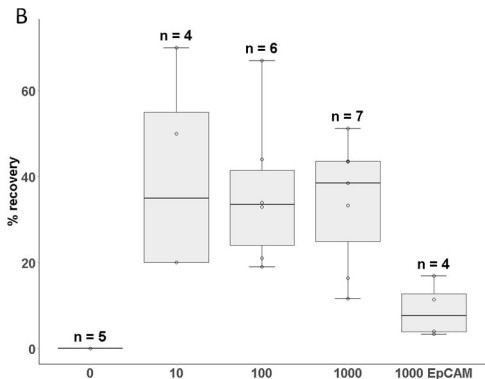

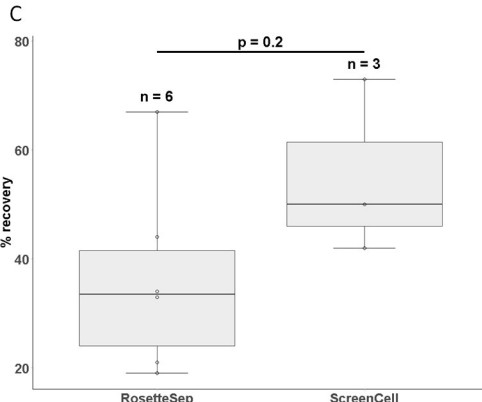

**Fig 2. Comparison of the various methods used for CTC isolation.** (A) Combined immunomagnetic isolation efficiency using Dynabeads® and EasySep™ compared to RosetteSep™ at 1000 cells, p = 0.004 between Dynabeads® and RosetteSep™ (B) The isolation efficiency of RosetteSep™ at 1000, 100, 10 cells and further enrichment with EasySep™ EpCAM at 1000 cells. (C) The data comparison of non-immunomagnetic isolations using RosetteSep™ and ScreenCell® filters at 100 cells, p = 0.18.

(n = 4) cells gave mean recoveries of 36% ± 18% and 40% ± 24%, respectively (Fig 2B). Individual cell count data is provided as supporting information (S2 Table). Since leukocyte contamination can still be variable with this method, we coupled the RosetteSep™ isolation with the EasySep™ Human EpCAM Positive Selection Kit (n = 4) to determine if enrichment could be further improved. However the mean recovery of the spike-in cells was 9% ± 6%, which was no better than the combined immunomagnetic isolations.

The inter-assay variability in CTC isolation using the RosetteSep™ method and the inter-observer variability in the enumeration by ICC (Table 1) was also evaluated. One-hundred or ten MDA-MB-231 cells were spiked into 5 ml of blood. Isolations of spiked cells were performed by three operators and ICCs were performed by two observers. No CTC were detected when no cells were spiked in. Even though the factorial ANOVA analyses showed no significant differences in the overall inter-operator variability (p = 0.06), when cell counts are low (< 10 cells) the inter-assay variability becomes significant (p = 0.006). On the other hand, no significant differences were found between the observers (p = 0.51) (Table 1) even in the presence of low cell counts. The RosetteSep™ method was subsequently used for CTC isolations from the cohort of metastatic breast cancer patients by operators 1 and 2 for isolations, ICCs and enumerations.

The ScreenCell® Cyto filter device provided a rapid isolation of CTCs from 3 mL of blood with only a single wash step. These devices provided reduced variability compared to the RosetteSep™ system (RosetteSep™ %CV = 49 vs ScreenCell® Cyto %CV = 29), presumably due to the decreased number of manipulation steps. In the spiking experiments of 100 cells (n = 3), the recovery using ScreenCell® Cyto was 55% ± 16% and the recovery of the RosetteSep™ system was 36% ± 18%; however, the difference was not significant (p = 0.18; Fig 2C).

The time efficiency and ease of use of the RosetteSep™ and the ScreenCell® filters (Table 2) were compared. The hands-on-time for the ScreenCell® filters for CTC isolation was

**Table 1. Inter-assay and inter-observer variability using RosetteSep™ for CTC isolation and ICCs at 100 and 10 cells.**

**Inter-Assay Operator Variability**

| Run | 1 | 2 | 3 | Mean | STD | % CV |
|---|---|---|---|---|---|---|
| **Operator 1** | | | | | | |
| **10 cells** | 1 | 0 | 0 | 0.3 | 0.6 | 173 |
| **100 cells** | 25 | 7 | 11 | 14 | 9.5 | 66 |
| **Operator 2** | | | | | | |
| **10 cells** | 4 | 2 | 1 | 2.3 | 1.5 | 66 |
| **100 cells** | 32 | 10 | 26 | 23 | 11 | 50 |
| **Operator 3** | | | | | | |
| **10 cells** | 1 | 0 | 3 | 1.3 | 1.5 | 115 |
| **100 cells** | 22 | 17 | 13 | 17 | 4.5 | 26 |

**Inter-Assay Observer Variability**

| Run | | 1 | | | 2 | | | 3 | |
|---|---|---|---|---|---|---|---|---|---|
| **Operator** | 1 | 2 | 3 | 1 | 2 | 3 | 1 | 2 | 3 |
| **10 cells** | | | | | | | | | |
| **Observer 1** | 1 | 4 | 1 | 0 | 2 | 0 | 0 | 1 | 3 |
| **Observer 2** | 0 | 4 | 1 | 0 | 2 | 0 | 0 | 4 | 6 |
| **100 cells** | | | | | | | | | |
| **Observer 1** | 25 | 32 | 22 | 7 | 10 | 17 | 11 | 26 | 13 |
| **Observer 2** | 22 | 37 | 28 | 6 | 11 | 15 | 18 | 30 | 15 |

STD: Standard deviation; CV: Coefficient of variation

**Table 2. Comparison of the time required to analyze CTCs using RosetteSep™ and ScreenCell®.**

| | RosetteSep™ | | | ScreenCell® | | |
|---|---|---|---|---|---|---|
| | **Protocol** | **Skill/ complexity** | **Time** | **Protocol** | **Skill/ complexity** | **Time** |
| Sample Preparation | • Incubation with antibody cocktail | Moderate | ~ 25 min | • Buffer preparation, dilution of blood and incubation | Low | ~ 10 min |
| | • Sample dilution and layering onto Ficoll-Paque | | **Hands on time:** 5min | | | **Hands on time:** 2min |
| CTC isolation | • Ficoll-Paque gradient centrifugation | Moderate | ~45–50 min | • Diluted blood filtration | Low | ~ 5 min |
| | • CTC collection | | | | | **Hands on time:** 5 min |
| | • Cell washing | | | | | |
| | • Cytospin preparation | | **Hands on time:** 20 min | | | |
| **Total hands on time** | | | **25 min** | | | **7 min** |
| **Total time for CTC isolation** | | | **1h 15 min** | | | **15 min** |
| Immunocytochemistry | • Simple ICC | Low | ~3 h | • Antigen retrieval | Low | ~4 h |
| | | | | • Permeabilization | | |
| | | | | • ICC | | |
| **Total time** | | | **4h 15 min** | | | **4h 15 min** |
| Imaging | • Variable depending on sample | High | Variable | • Variable due to nature of the filter | High | Variable |

approximately 7 min compared to 25 minutes of hands-on-time for the RosetteSep™ where slightly more skill is also required for the CTC isolation. The overall time for the CTC isolation only using the ScreenCell® was 15 min compared to the 1 h 15 min for the RosetteSep™ method. However, because cells prepared on the ScreenCell® Cyto filters are fixed in formaldehyde, an antigen retrieval step along with cell permeabilization was required for ICC. Finally, both methods require a trained and experienced observer for identifying and imaging CTCs. On the criteria of a CTC as CD45$^-$/pan-CK/CK-7$^+$/DAPI$^+$, all methods tested demonstrated 100% specificity as un-spiked controls had no positive cells.

## ICC staining with CD45, pan-cytokeratin and vimentin for identification and enumeration

A 3-colour fluorescence combination was used for the ICCs to distinguish tumor cells from contaminating leukocytes. The pan cytokeratin (CK 1–8, 10, 13–16, 18, 19) and cytokeratin-7 combination was chosen to encompass the diverse expression of CK by epithelial cells in their various stages of differentiation [19, 20]. Cells were identified as CTCs based on the criteria mentioned previously. There was very little ambiguity when applying these criteria to cultured cells (Fig 3). While CD45 is uniquely associated with leukocytes, they sometimes exhibit modest non-specific staining with cytokeratins (CD45$^+$/pan-CK$^+$). Hence, it was critical that all cells counted as CTCs do not have CD45 staining or a lobed nucleus (characteristic of leukocytes).

## CTC enrichment in metastatic breast cancer patients

Based on the various advantages and recovery rates of the different methodologies, the RosetteSep™ method and, when it became available, the ScreenCell® method were used to isolate CTCs from patient samples. Twenty-three metastatic breast cancer patients between the ages of 41–78 (median = 58 years) were enrolled and screened for CTCs (22 females and 1 male; Table 3). Eighty-three percent of the patients had ductal histology while 13% were lobular and

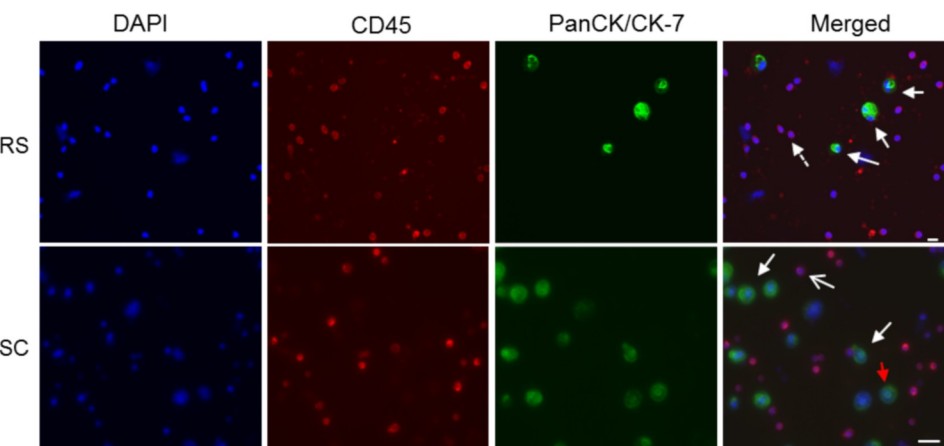

**Fig 3. Representative images of controls using spiked MDA-MB-231 cells into whole blood after CTC enrichment and immunocytochemistry.** Top panel: Cytospin preparations from CTC enrichment using the RosetteSep™ method. One hundred MDA-MD-231 cells were spiked into 5 mL of whole blood. Positive cells for cytokeratins are coloured green (white arrow), leukocyte positive for CD45 are coloured red (dashed arrow) and the nuclei stained by DAPI are shown in blue. Images were captured under 200X magnification. Bottom panel: CTC enrichment by ScreenCell® Cyto filters. One hundred cultured cells were spiked into 3 mL of whole blood. Cells that are CD45-, pan-CK/CK-7+ and with an intact nucleus are considered positive. Filter pores are visually distinct and a CTC caught atop a filter pore is indicated by the red arrow. A leukocyte caught in the filter pore is shown with an opened white arrow. Images were captured under 400X magnification. Scale bar represents 20 μm. RS: RosetteSep™; SC: ScreenCell®.

one was unknown (4%). Eight (35%) of the patients were of the basal molecular subtype, while 12 (52%) were of the luminal molecular subtype and 3 (13%) were Her2-positive. At the time of enrolment, all metastatic patients presented with at least one visceral site of metastatic spread (most were in lung and liver). Eighteen patients (78%) had relapsed or had progressive disease and 5 (22%) were newly diagnosed with metastatic disease. Four of the patients had multiple sampling resulting in 32 separate isolations.

Twenty four CTC isolations (75%) were performed using only the RosetteSep™ method, two isolations were performed using only the ScreenCell® method while 6 CTC isolations (18%) were performed using both the RosetteSep™ and ScreenCell® filters. Of the 32 isolations, 18 (56%) were positive for CTCs and 12 (38%) were negative while 2 (6%) were inconclusive due to poor staining on the cytospin slides. Fig 4 shows patient samples isolated with either RosetteSep™ or ScreenCell® filters or with both. The RosetteSep™ method (n = 28) was able to detect CTCs in 54% of samples with an average of 0.55 CTCs/mL of blood while the Screen-Cell® Cyto filters (n = 8) detected CTCs in 75% of the patient samples at an average of 4.2 CTCs/mL (Fig 5A). Of the 6 isolations using both methods, 5 (83%) of the isolations had CTC counts by both methods with an average of 0.2 CTCs/mL by RosetteSep™ compared to 0.6 CTCs/mL by ScreenCell® (Fig 5B). In one sample (17%), CTC were found by ScreenCell® isolation (1 CTC/mL) but not with RosetteSep™. Finally, 2 CTC isolations were (6.25%) performed with only the ScreenCell® filters, and CTCs were detected at 3.7 CTCs/mL and 26 CTCs/mL, respectively. The ScreenCell® filters detected 4.2 CTCs/mL compared to 0.55 CTCs/mL by RosetteSep™; however, the difference was not significant (p = 0.29).

In one of the samples where there was only sufficient blood for enrichment with Screen-Cell® filters, CTC clusters as well as single CTCs was observed when stained with both pan-CK/CK-7 and vimentin (Fig 6) in both ScreenCell® filters. The CTC clusters were panCK/CK7+ and CD45-. As both pan-CK/CK-7 and vimentin were used together with the same fluorophore, the possibility of the clusters being endothelial cells which are Vim+ cannot be discounted.

**Table 3. Patient clinical characteristics.**

| Characteristics | All patients (n = 23) |
|---|:---:|
| **Age** | |
| Median | 58 |
| Range | 43–78 |
| **Gender** | |
| Female | 22 (96%) |
| Male | 1 (4%) |
| **Histological findings** | |
| Ductal | 19 (83%) |
| Lobular | 3 (13%) |
| Unknown | 1 (4%) |
| **Number of metastatic sites** | |
| 1 | 10 (43%) |
| 2 | 9 (40%) |
| >3 | 4 (17%) |
| **ER** | |
| Positive | 12 (52%) |
| Negative | 11 (48%) |
| Unknown | 0 |
| **PR** | |
| Positive | 4 (17%) |
| Negative | 15 (66%) |
| Unknown | 4 (17%) |
| **Her2** | |
| Positive | 4 (17%) |
| Negative | 18 (79%) |
| Unknown | 1 (4%) |
| **Lines of therapy** | |
| 1 | 8 (35%) |
| 2 | 10 (43%) |
| >3 | 5 (22%) |
| **CTCs** | |
| Yes | 14 (61%) |
| No | 9 (39%) |

Therapy lines are either chemotherapies, anti-hormone therapies or other anti-cancer treatment.

## Correlation of CTCs to systemic treatment

The association of CTCs to systemic treatment and disease activity was examined in the two patients with more than two sampling times (Fig 7). In one patient (Patient 2), CTCs were measured at five time points collected between September, 2010 and January, 2012. This (patient 2) was a 42 year old woman who was initially treated for locally advanced ER(-), PR (-), HER2(-) breast cancer, with early recurrent metastatic breast cancer post mastectomy. Her systemic therapies included foretinib, as part of a clinical trial NCT 01147484 (https://clinicaltrials.gov), cisplatin and gemcitabine, and lastly nanoparticle albumin-bound paclitaxel. Her first blood sample, taken during disease progression had 2 CTCs (0.38 CTC/mL) by the RosetteSep™ method as did her second blood sampling (0.36 CTC/mL). Her third sampling; however, had no CTCs but on the fourth sampling, where clinical disease was progressing, 2

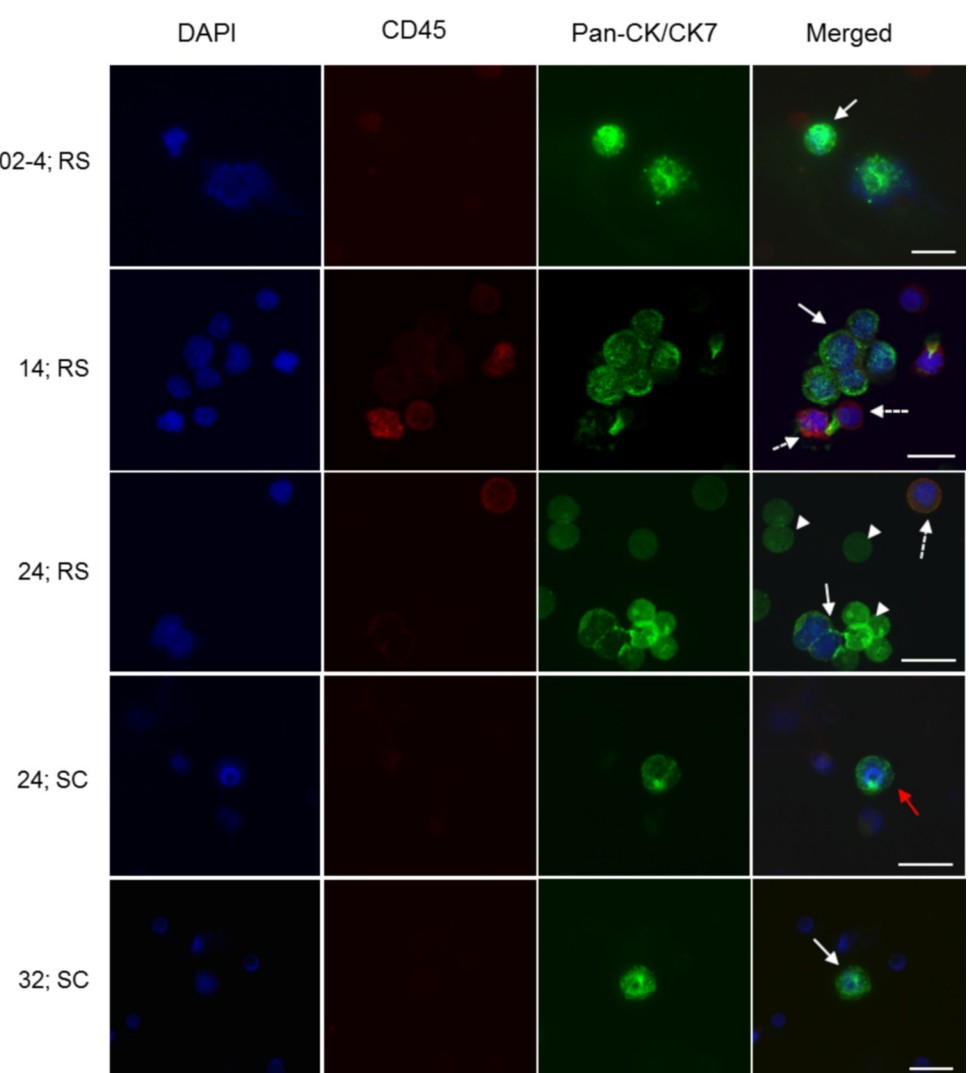

**Fig 4. Representative images of CTC enrichment and ICCs from four metastatic breast cancer patients.** CTCs were isolated using either the RosetteSep™ (RS) or ScreenCell® (SC) or with both. Numbers of the left denotes patient ID. CTCs and circulating tumor clusters showing pan-CK/CK-7$^+$ (green; white arrow) with an intact nucleus (DAPI$^+$, in blue) are CD45$^-$. Leukocytes are CD45$^+$ (dashed arrow) with no CK staining. The arrow head points to un-lysed residual red blood cells from the RosetteSep™ isolation. The red arrow points to a CTCs caught atop a filter pore (8 μm). Images were captured under 400X magnification. Scale bar represents 20 μm. RS: RosetteSep™; SC: ScreenCell®.

CTCs (0.40 CTC/mL) were detected by RosetteSep™. No CTCs were detected on the fifth sampling; however, since false negatives are common, we cannot rule out that the possibility that the apparent correlation with the imaging data is an artifact. Fig 7A shows the radiographic correlative findings of her disease course as assessed by computed tomography (CT) scans during treatment with cisplatin and gemcitabine.

Patient 23 was a 65 year old woman diagnosed with a phenotypically similar ER(-), PR(-), HER2(-) recurrent metastatic breast cancer. She had three blood samples drawn between May 2011 and May 2012, at which time she was treated with capecitabine. Prior to that, she was enrolled on the THYME clinical trial NCT00900627 (https://clinicaltrials.gov/), and after capecitabine she was treated with doxorubicin and dexrezoxane. No CTCs were detected on her

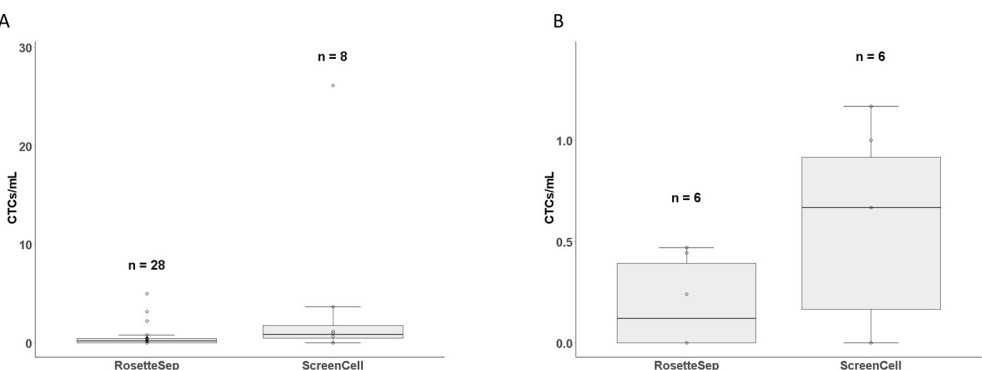

**Fig 5. CTC isolations from metastatic breast cancer patients.** (A) Isolation of CTCs by RosetteSep™ and ScreenCell® methods of all patient samples (p = 0.29). (B) number of CTCs isolated by both the RosetteSep™ and ScreenCell® methods from 6 metastatic breast cancer patients (p = 0.13).

first sampling in May 2011 but one CTC (0.44 CTC/mL) was detected in her second sampling as clinical disease progressed (September 2011) and before treatment (Fig 7B). Further, no CTCs were found in the third sampling.

## Discussion

CTC isolation remains a challenge due to the scarcity and heterogeneity of these cells in blood. Ideally, a combination of high sensitivity immunoaffinity-based positive selection combined with negative depletion of unwanted cells would provide pure CTC for downstream analysis.

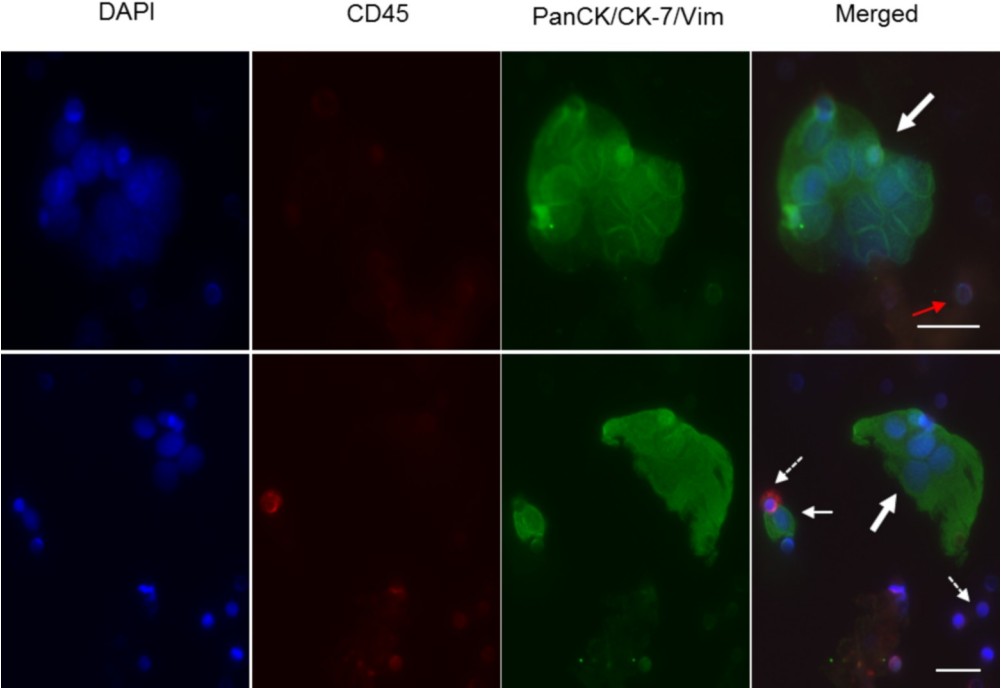

**Fig 6. Patient 38 with circulating tumour clusters isolated by the ScreenCell® Cyto kit.** ICCs were performed using anti-pan-CK/CK-7/vim and CD45. Cell nuclei are counterstained with DAPI. Circulating tumor clusters (bold arrow) showing pan-CK/CK-7$^+$/vim$^+$ but CD45$^-$. CD45$^+$ leukocytes are sometimes CK$^+$. Cells caught in the filter pores (red arrow). Images were captured under 400X magnification. Scale bar represents 20 μm.

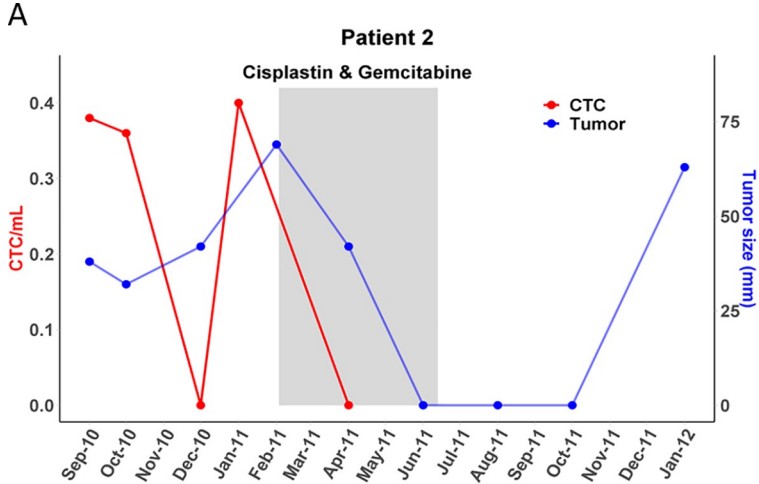

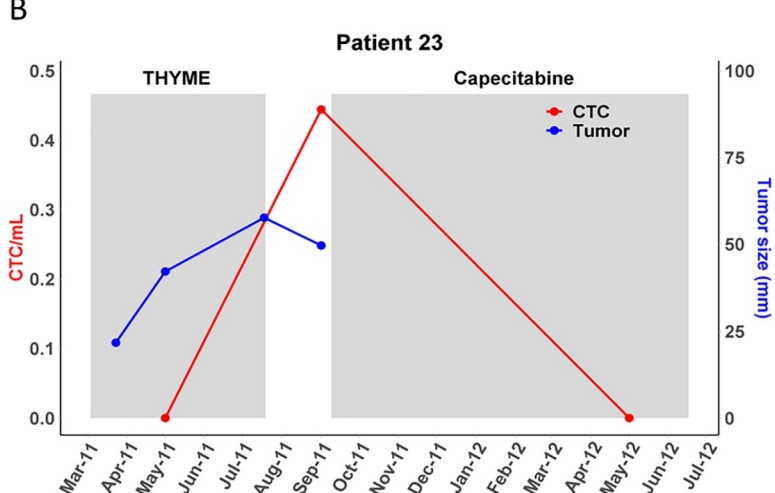

**Fig 7. Serial CTC and CT scan monitoring during systemic treatment.** Coloured boxes indicate the duration of therapy. CTC detection may be used to indicate disease progression. CT: computed tomography.

However, in our hands, the dual approach resulted in the lowest recovery rates (Fig 2A) compared to the other methods tested (Fig 2C). This is likely a result of the multiple washing and sample transfer steps making this impractical for use with patient samples. Further Dynabeads® bound cells are difficult to enumerate by ICC owing to the overwhelming amount of beads, the auto-fluorescence of the beads, and inefficient labelling by antibodies of cells when bead bound. By contrast, the smaller EasySep™ nanoparticles did not interfere with downstream ICC processing but their poor recovery rates at low spike levels preclude their use in clinical samples.

The RosetteSep™ isolation method does not rely on the antigen expression on CTCs but instead changes the density of the unwanted cells through the binding of antibodies and removal by density gradient centrifugation. CTCs isolated with this method are amenable to downstream processing such as nucleic acid extraction, cell culture, flow cytometry and ICC. Cytospin preparations of isolated cells can be stored for batch processing. However, this method can be time consuming and the manual processing limits the robustness and reproducibility where variable recovery may have a significant impact when cell numbers are low, as

observed in the inter-operator and inter-observer variability assessment. An improvement to this may be to combine the RosetteSep™ with the SepMate™-50 tubes for the Ficoll-gradient centrifugation. The SepMate™-50 was only available later in the study and to keep with consistency these tubes were not used for CTC isolations. The capture efficiency for the RosetteSep™ in spiking experiments was approximately 40% and detected CTCs in 54% (15/28) of MBC patients. He et al. [21] reported a recovery efficiency of 62.5% by RosetteSep™ in their spiked samples and was able to detect CTCs in 90% of metastatic epithelial ovarian cancer and 77% of prostate cancer by flow cytometry. A study by Kulasinghe et al. [22] showed a CTC detection rate of 70% in spike-in and 64% by RosetteSep™ in advanced stages of head and neck cancer when enumerated by ICC. Maertens et al., [23] on the other hand, had similar recovery rates of 40% with spike-in experiments using renal cell carcinoma cell lines by ICC. Aside from inter-laboratory variation, another contributing factor to lower recovery rates is cell loss during the cytospin preparations and ICC [24].

The ScreenCell® method gave the highest mean recovery of 55% for the spike-in experiments and detected CTCs in 75% (6/8) of the MBC patients. A similar study [25] using the ScreenCell® Cyto filters had a mean detection rate of 77% in lung cancer. This method for CTC isolation is also non-antigen dependent, is simpler than the RosetteSep™ and rapid with captured cells being amenable to different downstream applications including cell culture. The filters with the captured cells can also be stored for ICC batch processing. The ScreenCell® filters were able to capture more cells per mL than the RosetteSep™ system and this could be attributed to the fewer manipulation steps before ICC. However, because of the size heterogeneity among CTCs, smaller CTCs may be lost or be lodged within the ScreenCell® filter pores. This, along with the fact that the filter does not always sit flat, compounds the challenges in enumeration by imaging. The limited volume of blood that can be processed using the Screen-Cell® is likely the main limit to the sensitivity of this approach. Regardless, a few studies [25, 26] have shown its higher capture efficiency than the CellSearch® system. Furthermore, several studies have shown [20, 27, 28] that EpCAM based enrichment such as the CellSearch® system is less sensitive compared to those that are independent of surface antigens such as those that employ the filtration method [24, 29, 30]. The changing antigen profiles of CTCs undergoing EMT would suggest a strategy that is not entirely reliant on the use of EpCAM but rather employs multiple strategies, such as antibody mixtures [31, 32] or multiple methods [20, 33]. Both the RosetteSep™ and ScreenCell® methods were able to capture CTC clusters as also reported by others [22]. These are usually associated with poor outcome as they have a greater metastatic potential [34].

A major limitation of these two platforms is the manual enumeration of cells by ICC. Visual identification of CTCs is tedious, requires a higher level of training and can be subjective. Therefore, there is a critical need for technologies to automate and standardize optical detection of CTC in order for these techniques to become more robust and more amenable to routine use [35]. Nevertheless, we demonstrate that CTC analysis using RosetteSep™ and ScreenCell® is sensitive, compared to EasySep™ and Dynabeads®, and that the technique can be used for detection of CTC in metastatic breast cancer patients. In order to improve patient outcomes, further research is needed to link CTC phenotype to predictive biomarkers that could be used for treatment selection and optimization.

## Conclusions

The RosetteSep™ and ScreenCell® systems are able to isolate CTCs and CTC clusters from metastatic breast cancer patients. These techniques also provide adequate sensitivity at levels as low as 2 CTCs/mL, which is not the case for the Dynabead® and EasySep™ techniques.

The independence of EpCAM expression and simple operation using standard laboratory equipment render the RosetteSep™ and ScreenCell® techniques amenable to routine analysis in clinical settings. The results presented here warrant further investigation in larger, prospective studies in order to further explore the utility of CTC analysis as an actionable assay in clinical settings.

## Supporting information

**S1 Table.**
(PDF)

**S2 Table.**
(PDF)

**S3 Table.**
(PDF)

## Acknowledgments

We thank Ken Chisholm and Jeff Gallant for their technical assistance for CTC isolations and Andrew Leslie for the statistical analysis.

## Author Contributions

**Conceptualization:** Arik Drucker, Evelyn M. Teh, Daniel Rayson, Susan Douglas, Devanand M. Pinto.

**Data curation:** Arik Drucker, Evelyn M. Teh, Ripsik Kostyleva, Susan Douglas, Devanand M. Pinto.

**Formal analysis:** Evelyn M. Teh.

**Funding acquisition:** Arik Drucker, Devanand M. Pinto.

**Investigation:** Evelyn M. Teh, Devanand M. Pinto.

**Methodology:** Arik Drucker, Evelyn M. Teh, Susan Douglas, Devanand M. Pinto.

**Project administration:** Arik Drucker, Devanand M. Pinto.

**Supervision:** Arik Drucker.

**Visualization:** Daniel Rayson, Devanand M. Pinto.

**Writing – original draft:** Arik Drucker, Evelyn M. Teh, Devanand M. Pinto.

**Writing – review & editing:** Arik Drucker, Evelyn M. Teh, Ripsik Kostyleva, Daniel Rayson, Susan Douglas, Devanand M. Pinto.

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
