## [Decision Letter · Decision Letter 0]

14 Oct 2019

PONE-D-19-23597

Comparative Performance of Different Methods for Circulating Tumor Cell Enrichment in Metastatic Breast Cancer Patients

PLOS ONE

Dear Dr. Drucker,

Thank you for submitting your manuscript to PLOS ONE. After careful consideration, we feel that it has merit but does not fully meet PLOS ONE’s publication criteria as it currently stands. Therefore, we invite you to submit a revised version of the manuscript that addresses the points raised during the review process. For successful revision of the study, it is expected that all concerns by the Reviewers are adequately addressed, including improving clarity for the experimental design and the results, explaining clearly the contribution of this work in relation to more recent published literature and providing additional data and analyses as required.

We would appreciate receiving your revised manuscript by Nov 28 2019 11:59PM. To enhance the reproducibility of your results, we recommend that if applicable you deposit your laboratory protocols in protocols.io, where a protocol can be assigned its own identifier (DOI) such that it can be cited independently in the future. For instructions see: http://journals.plos.org/plosone/s/submission-guidelines#loc-laboratory-protocols

We look forward to receiving your revised manuscript.

Kind regards,

Sophia N Karagiannis, BA, MS, PhD

Academic Editor

PLOS ONE

Journal Requirements:

2. At this time, we ask that you please add scale bars to the microscopy images provided in the manuscript and add the scale represented in the appropriate figure legend. Thank you for your attention to this request.

3. In your Methods section, please provide additional information in the your Methods section about the dates (month/year) of enrolment of the breast cancer patients used in this study.

Additionally, please provide additional details regarding the breast cancer patients consent. In the ethics statement in the Methods and online submission information, please ensure that you have specified (1) whether consent was informed and (2) what type you obtained (for instance, written or verbal, and if verbal, how it was documented and witnessed). If your study included minors, state whether you obtained consent from parents or guardians. If the need for consent was waived, please ensure that you have discussed whether all data were fully anonymized before you accessed them and/or whether the IRB or ethics committee waived the requirement for informed consent.”

4. We suggest you thoroughly copyedit your abstract for language usage, spelling, and grammar. If you do not know anyone who can help you do this, you may wish to consider employing a professional scientific editing service.  

7.  Thank you for stating the following in the Acknowledgments Section of your manuscript:

This study was supported in part by the Capital Health Research Fund Category 2 Grant and the National Research Council of Canada Atlantic Initiate Fund.

A.D - University Internal Medicine Research Fund, CDHA Category 2 Fund, Pfizer Oncology IIR

D.P., S.D., E.T.- National Research Council Atlantic Initiative

8. We note that you received funding from a commercial source: Pfizer Oncology IIR

Reviewers' comments:

Reviewer's Responses to Questions

**Comments to the Author**

1. Is the manuscript technically sound, and do the data support the conclusions?

Reviewer #1: Yes

Reviewer #2: No

2. Has the statistical analysis been performed appropriately and rigorously? 

Reviewer #1: Yes

Reviewer #2: No

3. Have the authors made all data underlying the findings in their manuscript fully available?

Reviewer #1: Yes

Reviewer #2: No

4. Is the manuscript presented in an intelligible fashion and written in standard English?

Reviewer #1: Yes

Reviewer #2: Yes

5. Review Comments to the Author

Reviewer #1: The manuscript(MS#PONE-D-19-23597) mainly compared the potential of four methods in the CTCs identification in order to find a simpler and faster method. The methods provided a potential for clinical practice. Actually, several methods have been reported to determine CTCs for cancer prognosis and diagnosis. Therefore, what is the novelty of the developed method? Some other issues should be addressed.

1. Two cell lines (MDA-MB-231 and MCF-7) were selected in this study, but why was only MDA-MB-231 used in “Spiking experiments with healthy controls”?

2. How to ensure that the cells isolated are the target tumor cells?

3. Please provided the detailed figure legends. All the figures are not unambiguous, please make all the figures clear.

4. Please explain why there are different experiment times in different tests, as shown in Figure 2B.

5. In Table 2, the total time of ScreenCell should be ‘4h 15min’, but not ‘4h 7min, as well as Line 259.

6. Please make a short conclusion for this research.

Reviewer #2: Drucker et al evaluated four techniques for isolation and analysis of CTCs and found that these systems had comparable performance when tested using high concentrations of cultured cell spike-in models. Two techniques fared better at lower concentrations and were tested using clinical blood samples from patients with metastatic breast cancer.

The experimental design is not clear, and the results are difficult to follow. The goal of comparing different CTC techniques is not novel. The authors could have designed a novel approach to head-to-head testing of different platforms, but this was not the case.

Introduction:

Reference 6. There are more recent reviews on CTC isolation. Please update your literature citations.

Refence 10. Please provide an appropriate citation regarding low expression of EPCAM in certain tumor types, e.g., triple negative and NSCLC.

Line 62-64. A non-sequitur. It is not clear what that point of these sentences are. Are these the rationale why you did not benchmark your experiments with an FDA-cleared test and compared correlation with clinical outcomes?

This head-to-head comparison study is not unique, please explain how your study is unique. Please also provide a supplementary table that summarizes the results from other related comparative studies.

Materials and Methods:

The number of spike-in cells is too high (10-100 fold higher) and does not reflect the clinical setting with very limited number of cells detected (approximately <1CTC/mL in metastatic breast cancer).

The experimental design is unclear. Please add a Figure 1b to illustrate study schema or the workflow of experiments. The current figure only describes the techniques.

Results:

Genomic testing of cells isolated in MBC patients were not performed so it is unclear whether cells detected by these methods are truly of malignant origin or may be just false positives. For examples, FISH test of CTCs from HER2+ patients could have been performed to address this question.

Please provide a supplementary table (CTC yields) that shows all the results of the experiments. This should be made available.

The X's before number of spiked cells in Figure 2B are R artifacts. Please clean up.

The negative x-axis range in Figure 5B is confusing.

Figure 7B (also not labelled as b) Patient 23 results: At first the results look interesting but upon closer examination two other points are 0. The lines seem to trick us that there is some correlation.

6. PLOS authors have the option to publish the peer review history of their article (what does this mean?). If published, this will include your full peer review and any attached files.

Reviewer #1: Yes: WEI-HUA HUANG from Central South University, China

Reviewer #2: No

---

## [Author Response · Author response to Decision Letter 0]

31 Dec 2019

Responses to reviewers are provided in the rebuttal letter, and changes to the manuscript have been made that address the concerns of reviewers.

---

## [Editor Report · Decision Letter 1]

27 Jan 2020

PONE-D-19-23597R1

Comparative performance of different methods for circulating tumor cell enrichment in metastatic breast cancer patients

PLOS ONE

Dear Dr. Drucker,

Thank you for submitting your manuscript to PLOS ONE. After careful consideration, we feel that it has merit but does not fully meet PLOS ONE’s publication criteria as it currently stands. Therefore, we invite you to submit a revised version of the manuscript that addresses the points raised during the review process.

a) In the Abstract:

- Please identify the marker used for cell isolationt.

- The final statement mentions the final experiment conducted but does not describe the findings: "These techniques were then applied to the isolation and analysis of circulating tumor cells blood drawn from metastatic breast cancer patients.". Provide a summary on the findings from these experiments.

b) In the Conclusion statement:

- Please correct the following sentence: "The results presented her warrant further investigation of in larger, prospective studies..."

We would appreciate receiving your revised manuscript by Mar 12 2020 11:59PM. To enhance the reproducibility of your results, we recommend that if applicable you deposit your laboratory protocols in protocols.io, where a protocol can be assigned its own identifier (DOI) such that it can be cited independently in the future. For instructions see: http://journals.plos.org/plosone/s/submission-guidelines#loc-laboratory-protocols

We look forward to receiving your revised manuscript.

Kind regards,

Sophia N Karagiannis, BA, MS, PhD

Academic Editor

PLOS ONE

---

## [Author Response · Author response to Decision Letter 1]

23 Jul 2020

Revisions are made and marked up in the abstract and typographical errors corrected, as requested by PLoS.

---

## [Editor Report · Decision Letter 2]

27 Jul 2020

Comparative performance of different methods for circulating tumor cell enrichment in metastatic breast cancer patients

PONE-D-19-23597R2

Dear Dr. Drucker,

We’re pleased to inform you that your manuscript has been judged scientifically suitable for publication and will be formally accepted for publication once it meets all outstanding technical requirements.

Kind regards,

Sophia N Karagiannis, BA, MS, PhD

Academic Editor

PLOS ONE
---

## [Editor Report · Acceptance letter]

4 Aug 2020

PONE-D-19-23597R2 

Comparative performance of different methods for circulating tumor cell enrichment in metastatic breast cancer patients 

Dear Dr. Drucker:

I'm pleased to inform you that your manuscript has been deemed suitable for publication in PLOS ONE. Congratulations! Your manuscript is now with our production department. 

Kind regards, 

on behalf of

Dr Sophia N Karagiannis 

Academic Editor

PLOS ONE